# Athletic Trainers’ Perceptions of and Experience with Social Determinants of Health

**DOI:** 10.3390/ijerph20085602

**Published:** 2023-04-21

**Authors:** Kelsey J. Picha, Cailee E. Welch Bacon, R. Curt Bay, Joy H. Lewis, Alison R. Snyder Valier

**Affiliations:** 1Department of Interdisciplinary Health Sciences, Arizona School of Health Sciences, A.T. Still University, Mesa, AZ 85206, USA; 2Department of Athletic Training, Arizona School of Health Sciences, A.T. Still University, Mesa, AZ 85206, USA; 3Department of Basic Science Education, School of Osteopathic Medicine in Arizona, A.T. Still University, Mesa, AZ 85206, USA; 4Department of Public Health, School of Osteopathic Medicine in Arizona, A.T. Still University, Mesa, AZ 85206, USA

**Keywords:** athletic healthcare, social factors, health inequalities, patient-centered care

## Abstract

The role that social determinants of health (SDHs) play in athletic healthcare is gaining attention, yet little is known about athletic trainers’ (ATs) perceptions of and encounters with the impact of SDHs. The purpose of this study was to evaluate ATs’ perceptions of various SDHs and their experience treating patients whose health and well-being were influenced by SDHs. This was a cross-sectional, web-based survey completed by 1694 ATs (completion rate = 92.6%; 61.1% female; age = 36.6 ± 10.8 years). The survey consisted of several multipart questions focusing on specific SDHs. Descriptive statistics were used to report frequencies and percentages. Results indicated widespread agreement that SDHs matter to patient health and are of concern in athletic healthcare. The SDHs that ATs most commonly reported encountering included lifestyle choices (n = 1306/1406; 93.0%), social support (n = 1185/1427; 83.0%), income (n = 1167/1502; 77.7%), and access to quality and timely healthcare (n = 1093/1420, 77.0%). The SDHs that ATs least commonly reported having experience with was governmental policy (n = 684/1411; 48%). The perceived importance of SDHs among ATs and their commonly reported experiences managing patient cases in which SDHs negatively influence patients’ health and healthcare suggest that efforts to assess these factors are needed so that strategies to address their influence on athletic healthcare can be identified.

## 1. Introduction

Clinicians in all facets of healthcare strive to provide the highest quality care to their patients, yet complex social factors outside the clinic may outweigh their best efforts. Social determinants of health (SDHs) have been estimated to influence patient health and well-being to a greater extent than direct healthcare affects patient outcomes [1,2,3,4]. Social determinants of health are “the conditions in which we live, learn, work, play and age” [5], and are determined by the division, often unequal, of resources, power, and money at all levels of government [1]. The Centers for Disease Control and Prevention categorizes SDHs into five categories: healthcare access and quality, education access and quality, economic stability, neighborhood and build environment, and social and community context [6]. Social determinants of health are interconnected, complex, and may have positive or negative influences on health. When SDHs negatively influence health, health inequalities arise. Addressing SDHs has become a high priority among healthcare communities [7,8].

Prior to the implementation of interventions to address SDHs, it is important to understand what healthcare providers know about SDHs and what they perceive to be relevant to providing care. Clinicians in other health professions have reported the impact of SDHs on their patients. Survey results of primary care physicians consistently report that social factors play an important role in the health outcomes of their patients [9]. However, most are not confident in their ability to address individual patient social needs, and identifying ways to include assessment of SDHs as part of patient care is a challenge. Understanding perceptions and experiences with SDHs should create a foundation of information that will be useful in supporting clinical decisions and intervention strategies for patients experiencing these negative factors. While the studies from physician practice are a start to building understanding, these data are limited to more traditional clinical practice settings. Other models of healthcare, such as those used in athletic healthcare, warrant study to learn more about the SDHs in those patient populations.

As healthcare providers who have frequent contact with patients, often in school settings, athletic trainers (ATs) are uniquely positioned to identify and address SDHs. Within athletic patient populations, the recognition of SDHs has recently gained momentum [10,11,12]. Athletic trainers are healthcare providers who are responsible for emergency care, prevention, clinical examination and diagnosis, rehabilitation and treatment, and patient education of injury and illness and are well positioned to identify the influence of SDHs among their patient populations and assist in addressing negatively influential SDHs at the individual level. In some instances, such as the secondary school setting, the AT may be the only healthcare provider the patient sees [13]. Their interactions may occur on a daily basis, quite different from the predominant healthcare model in which care providers address an immediate health concern during one office visit. Additionally, while ATs care for a variety of types of patients, traditional settings emphasize high-functioning patient populations, such as sports participants, industrial personnel, and tactical athletes. The ability of an AT to provide frequent preventative and restorative care is unique. These features present opportunities to promote awareness, assessment, and advocacy to address SDHs in athletic healthcare.

While ATs routinely mitigate deficits in health literacy and transportation that may be associated with a history of limited access to healthcare [10], they report minimal to moderate knowledge about, familiarity with, and comfort in identifying SDHs [14]. ATs’ perceptions of the influence of SDHs on patient health and their experience managing patient cases where SDHs may influence outcomes have not been explored. A greater understanding of the SDH factors that impact patients will provide a foundation to support clinical decisions and treatment strategies. Therefore, the purpose of the current study was to determine whether ATs perceive SDHs to be impactful to patient lives and to explore whether they have managed a patient in which specific SDHs were salient.

## 2. Materials and Methods

### 2.1. Design and Participants

A cross-sectional, web-based survey was used to assess ATs’ perceptions of and experiences with SDHs in athletic healthcare. The University’s Institutional Review Board deemed this study exempt. A total of 17,000 ATs who were certified and members of the National Athletic Trainers’ Association (NATA) in good standing were invited to participate. Athletic trainers with various education levels in different settings and positions were included to ensure that the results reflected the NATA membership.

### 2.2. Instrumentation

The Athletic Trainers’ Perceptions of Social Determinants of Health Survey (AT-SDH) [14] was used to evaluate ATs’ perceptions of and experiences with SDHs. This survey was adapted from two previously validated surveys [15,16], reviewed by content experts, and pilot tested by individuals who fit the inclusion criteria for this study. Three questions were asked related to each of 10 SDHs (Table 1): (1) AT agreement as to whether a certain SDH influences patient health and well-being, (2) AT agreement as to whether a certain SDH is of concern in athletic healthcare, and (3) whether the AT has managed a patient case where a certain SDH negatively influenced a patient’s health and well-being. The agreement questions were rated on a 4-point Likert scale (i.e., strongly agree, agree, disagree, and strongly disagree), and the management question options were yes, no, or unsure. The survey was designed to take 10–15 min to complete and was hosted in the Qualtrics platform (Qualtrics, Inc., Provo, UT, USA).

### 2.3. Procedures

During the Fall of 2019, a convenience sample of 17,000 ATs was sent an email with the survey link through the NATA Survey Research Service. The email included study details, such as the purpose of the study, members of the research team, and survey information. Athletic trainer consent was assumed through voluntary completion of the survey. Weekly reminder emails were sent to those who had not yet completed the survey. The survey was available for a 4-week period.

### 2.4. Data Analysis

Descriptive statistics were used to characterize the AT data. A composite experience score was calculated by adding the number of ‘yes’ responses to the “Have you ever managed a patient case in which (insert SDH) negatively influenced his/her health and well-being?” A “yes” answer was awarded 2 points, an “unsure” answer was awarded 1 point, and a “no” answer was awarded 0 points. Composite experience scores ranged from 0 to 20; a score of 0 indicated no experience managing patient cases that involved the SDHs, while a score of 20 indicated the greatest experience. Statistical analyses were performed in SPSS version 26 (version 20.0; IBM Corp., Armonk, NY, USA).

## 3. Results

A total of 1829 of 17,000 ATs accessed the survey (access rate = 10.8%), and 1694 completed the survey (completion rate = 92.6%). Athletic trainer respondent demographics have been published elsewhere [14]. Most ATs were female (n = 856; 61.14%), averaged 37 ± 11.3 years of age, and had been practicing athletic training for 15.2 ± 10.6 years.

### 3.1. Perceptions of Social Determinants of Health

The majority of ATs agreed or strongly agreed that SDHs influence health and well-being (Figure 1) and, specifically, are of concern in athletic healthcare (Figure 2). The top three SDHs influencing a person’s health and well-being that ATs strongly agreed on were a person’s lifestyle choices (n = 1228/1406; 87.3%), access to quality and timely healthcare services (n = 1094/1422; 76.9%), and social support (n = 1065/1426; 74.7%). Conversely, the SDH ATs selected ‘strongly agree’ about the least was transportation (n = 554/1403; 39.5%). In athletic healthcare specifically, the top three SDHs ATs strongly agreed were a concern were lifestyle choices (n = 1024/1407; 72.8%), access to quality and timely healthcare services (n = 820/1419; 57.8%), and social support (n = 734/1427; 51.4%). Conversely, the SDH ATs selected ‘strongly agree’ about the least was education (n = 294/1475; 19.9%).

### 3.2. Experience with Social Determinants of Health

Overall, ATs’ experience scores ranged from 0–18, with a mean score of 14.3 ± 3.8. Figure 3 displays ATs’ experiences with each of the SDHs. The most commonly reported SDHs ATs have experience managing include lifestyle choices (n = 1306/1406; 93.0%), social support (n = 1185/1427; 83.0%), income (n = 1167/1502; 77.7%), and access to quality and timely healthcare (n = 1093/1420, 77.0%). The least commonly reported SDHs ATs have experience with were governmental policy (n = 684/1411; 48%), education (n = 884/1472; 60%), and employment (n = 903/1450; 62%). Governmental policy (n = 385/1411; 27%), early childhood experiences (n = 296/1414; 21%), and education (n = 264/1472; 18%%) were the most common SDHs that ATs were “unsure” if they have had experience with managing during a patient encounter.

## 4. Discussion

Factors outside of the care provided to patients in the clinic must be considered and addressed in order to improve overall patient health and well-being at the individual and population levels. This study was one of the first to explore ATs’ perceptions of and experiences managing cases where SDHs are perceived to be negatively influencing a patient’s health in athletic healthcare. Overall, ATs perceive SDHs to be influential to the health and well-being of patients and consider SDHs to be a concern within athletic healthcare. In addition, the majority of ATs reported having experience managing patient cases where SDHs were negatively influencing patient health. When seeking to make improvements or changes in behaviors, there is value in understanding peoples’ perceptions and experiences related to a concept or construct. Perceptions are often influenced by individuals’ experiences, and experiences can vary based on perceptions [17]. Therefore, it is not surprising that ATs in this study strongly agreed that lifestyle choices, social support, and access to healthcare influence patient health and well-being because many ATs also reported managing patient cases where SDHs were negatively influencing patient health. Athletic trainers are unique in that many interact with their patients often on a daily basis, giving them frequent opportunities to observe, assess, and have deeper conversations with their patients about SDHs.

The majority of ATs in this study agree or strongly agree that lifestyle factors influence patient health, and report lifestyle factors as the SDH they have the most experience in managing during patient interactions. Lifestyle choices were framed around patient diet, smoking status, consumption of alcohol, and exercise practices. Lifestyle choices have been found to account for 30% of the factors that contribute to overall health [18]. The literature demonstrates the impact smoking, alcohol consumption, and exercise have on many aspects of health [19]. Therefore, acknowledging, inquiring, and recognizing these as a healthcare provider is expected, and it is not surprising that ATs in this study report experience managing these SDHs. These findings provide additional opportunities to explore lifestyle choices patients make in athletic healthcare. Studies have suggested that most health behaviors and lifestyle choices are formed by economic and social influences [19,20]. One example is the increased use of social media and its impact on youths. Not only has social media influenced alcohol and cigarette use in adolescents [19,21], but a systematic review has also indicated social media is influencing food choices through the use of celebrity promotion and unhealthy food advertisements [22]. While we do not know the characteristics of patients reflected upon in this study, efforts to explore factors that promote healthy lifestyle choices, especially as a part of athletic healthcare, are needed. Further study into how ATs are managing patients who are negatively impacted by lifestyle choices is needed to explore helpful strategies to reduce these important SDHs.

Social support and its relationship to health and well-being have been well documented in athletic healthcare [23,24,25,26]. Social support, also referred to as social networks, is important to the health of people and populations [27]. Family, friends, and people in a person’s local environment are typically considered part of the social network [26], and in athletic healthcare, a network may also include teammates, coaches, and members of the medical staff. Positive and negative influence can result from social networks and their strong influence. Data from the current study suggest that ATs perceive the influence of social support in the care of patients, which is a positive. The majority of ATs in this study agreed or strongly agreed that social support is influential to patient health, and over 80% reported experience managing a patient case where social support was negatively impacting their patient’s health. These findings are reflective of Clement and Shannon [25], who studied injured athletes’ perceptions of social support provided by ATs, coaches, and teammates. Although their findings indicate the importance and value of social support provided by ATs, coaches, and teammates, athletes reported greater satisfaction with the availability and contribution of ATs’ social support compared to their teammates and coaches [25]. Similarly, athletes that reported satisfaction with the social support received from their AT were less likely to experience anxiety or depression symptoms when returning to play [28]. Athletic trainers are influential social support systems for their patients and recognize the value of social support when providing care. In addition to ATs providing social support to their patients, due to the close and regular interactions with their patients, it is also plausible that ATs observe and inquire about social support provided by others such as family, teammates, and coaches, which may further explain why ATs agree that social support is influential, whether negative or positive, to patients’ health and well-being, and is a concern in athletic healthcare.

Athletic trainers not only provide healthcare, but also provide patient education, promote patient health literacy by supporting their navigation of the health system, and advocate for patients [29,30]. When asked about the importance of quality and timely access to healthcare, most ATs agreed or strongly agreed that this SDH influences patient health and well-being and is a concern in athletic healthcare. Additionally, ATs’ experiences managing patient cases where access to quality and timely healthcare negatively impacted patient health were common. Access to healthcare is described as the ability to receive needed healthcare at the time healthcare is needed [31]. Healthcare access can be related to various services, such as prevention and treatment, across chronic and acute mental or physical health conditions. Factors such as insurance coverage, cost, transportation, and available providers can influence access to care [31]. Over the last decade, ATs have continued to gain a presence in secondary schools, ensuring athletes have quality and timely medical coverage at sporting events, both for practice and competition [32]. This increase in access to ATs has resulted in positive health outcomes [33]; yet, even when ATs are available on-site, they may not be able to overcome some of the obstacles when additional care is needed, such as when insurance and family finances limit care options. Athletic trainers in various settings may encounter these obstacles, despite their presence and ability to provide healthcare to their patients. With ATs’ perceived importance of and experience with this SDH, they are positioned to be advocates to promote access beyond the athletic healthcare environment.

Although the vast majority of ATs in this study agreed or strongly agreed that all SDHs were of concern in athletic healthcare, some SDHs, such as education, employment, transportation, and governmental policy, were among those with less agreement. Participation in athletics may naturally reduce some SDHs, and ATs may further contribute to reducing their negative impact [10]. Education and employment may be perceived to be less of a concern in athletic healthcare because often, the populations being served are enrolled in school, whether it be secondary school or college/university, or participating at an elite level, such as professional sports. Because these athletes are pursuing an education or participating through employment, ATs may not perceive these SDHs as a negative influence on patient health and well-being. Similarly, access to transportation may be viewed as less of an SDH concern in athletic populations where transportation is coordinated, such as in college/university athletics, or in situations where a medical provided is brought to patients through on-site healthcare visits [10]. Similarly, as part of routine care, ATs provide preventative, evaluative, and rehabilitative services to patients; therefore, patients may not need transportation to external sites because care is provided at the place of sport participation or employment. Overall, due to the natural roles and responsibilities of ATs and the settings in which they work, some SDHs may not be perceived to have a negative influence on patient health.

Not only was governmental policy agreed upon less often as a factor of concern in athletic healthcare, but it was also the SDH that ATs reported having the least experience with. Almost 52% of ATs in this study said “no” or that they were “unsure” if they had managed a patient case where governmental policy was negatively influencing their patient’s health. This finding is not surprising, as governmental policies and procedures are often upstream SDHs [34], ones that are more difficult to observe directly at a patient level. The definition of governmental policy in the survey included government policies and programs that affect health, social services, education, and the economy. Additionally, the data in this study were reported by ATs and may not be reflective of the communities that cannot afford an AT, and perhaps other social services or governmental programs would be more apparent in populations that do not have an AT. This specific SDH may be more difficult to observe within individual patients and may need to be studied further upstream at the population level.

Another SDH ATs more often reported “no” experience with or being “unsure” if they had experience with was early childhood experiences, defined in the survey as the type of parenting or upbringing and problems in the home. Although there is evidence to suggest early childhood experiences influence not only immediate health outcomes, but health outcomes throughout life [35], this SDH may be difficult to observe during patient interactions and not as easy to discuss with patients. However, knowledge of early childhood experiences may be an opportunity to influence longer-term health because, for example, children or adolescents that experience an adverse childhood event, such as neglect (direct) or parental divorce (indirect), have been found to suffer from more mental or physical health issues than those who do not [35,36,37]. This may be an area to further educate ATs and provide them with assessment tools and additional resources to better serve patients who may be negatively impacted by this SDH.

The SDHs are complex, sensitive, and sometimes hard to observe and discuss in clinical practice. Athletic trainers in this study are experiencing managing patient cases where some SDHs are present, and this recognition is positive. To produce improved patient outcomes, SDHs must be addressed as part of the healthcare experience, or the predominant influencing factors to long-term health are overlooked [38,39]. With ATs’ agreement that SDHs have an impact on patient health, experience managing patient cases where SDHs are present, and regular interactions with patients, the conditions are optimal to address SDHs in athletic healthcare. The addition of formally assessing SDHs may further solidify what the patient is experiencing, which would allow the AT to better provide direct care. Winkelmann et al. [12] found that in regard to public health tasks, ATs are more unsure of how to evaluate determinants of wellness and environmental factors than other public health tasks, and therefore, guidance from other professions may be helpful in identifying effective approaches starting with an assessment. The process of implementing an assessment strategy takes planning and preparation. Consideration of the patient population being served and what social factors may be common within that community is crucial [40]. Planning and preparation will allow for a customized assessment to address specific SDHs. Various screening tools exist to assess SDHs, some general and some more specific to one or two social factors [41,42]. One example is the American Academy of Family Physicians Social Needs Screening Tool [43], which is a 10-item screening that includes questions about housing, food, personal safety, and transportation. Bleacher et al. [44] have suggested that altering these screening tools to fit the needs of a specific patient population may be necessary and something ATs should consider doing. Athletic trainers are at the forefront of patient interaction and have the ability to begin to see trends in common SDHs and eventually become stronger advocates for their patients [45]. Future work in this area should focus on ATs observations and assessments of SDHs in athletic healthcare and, eventually, the development of a screening tool specific to athletic training practice, just as others have established for broader healthcare sectors.

### Limitations and Future Research

This study is not without limitations. While the access rate was low, 10.8% aligns with the average web-based survey access rates found in the athletic training literature [12,32,46]. Overall, the demographics are similar to the most recent Board of Certification AT demographic report (accessed on 17 April 2023 https://bocatc.org/system/document_versions/versions/293/original/at-demographics-20220414.pdf?1649950857) [14]. It is important to acknowledge that 65.2% of ATs who completed this survey were employed in either the secondary school, collegiate/university, or professional setting and thus are not representative of all ATs. As such, our findings may overestimate the perceptions about and experience with SDHs in the broader AT community. Second, the questions asked about the specific SDHs were purposefully broad, which limits the depth of what we know about the factors. For example, experience managing a patient case where access to quality and timely healthcare negatively influenced a patient’s health and well-being does not provide details of the nature of the encounter. More details about the experiences may provide insight into strategies and resources needed to support patients. Further investigation into ATs’ experiences could provide information to aid in the creation of specific resources to better manage patient cases impacted by SDHs.

## 5. Conclusions

In efforts to improve patient health and well-being, SDHs must be addressed as part of healthcare interactions. Athletic trainers in this study perceive SDHs as influential to patient health and that most are of concern in athletic healthcare and have experience managing them. As lifestyle choices, social support, and access to quality and timely care were most often experienced by ATs, efforts to better understand how these factors impact health are needed so strategies to address them can be identified. With awareness of SDHs and reported experience, ATs are in an ideal position to positively impact patient health and well-being by mitigating the negative influence of SDHs. Future research efforts are needed to explore effective assessment and intervention strategies aimed at reducing the negative impact of SDHs at the point of care.

## Figures and Tables

**Figure 1 ijerph-20-05602-f001:**
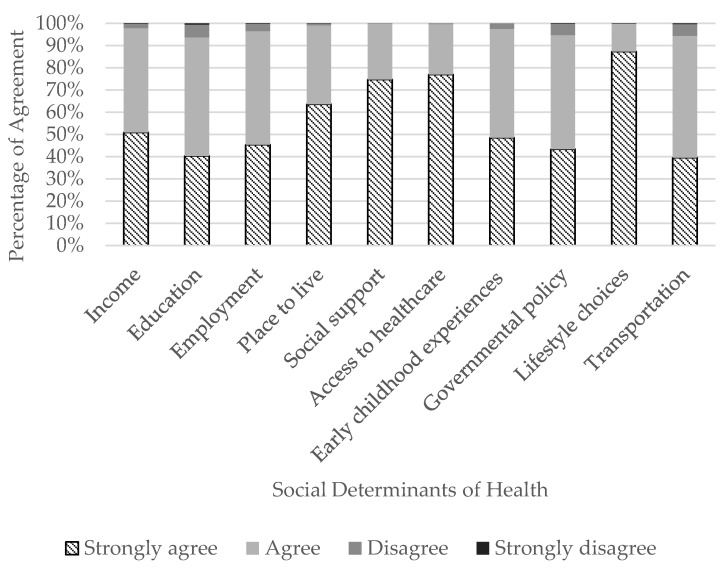
Athletic trainers’ perceptions of the influence social determinants of health have on people’s health.

**Figure 2 ijerph-20-05602-f002:**
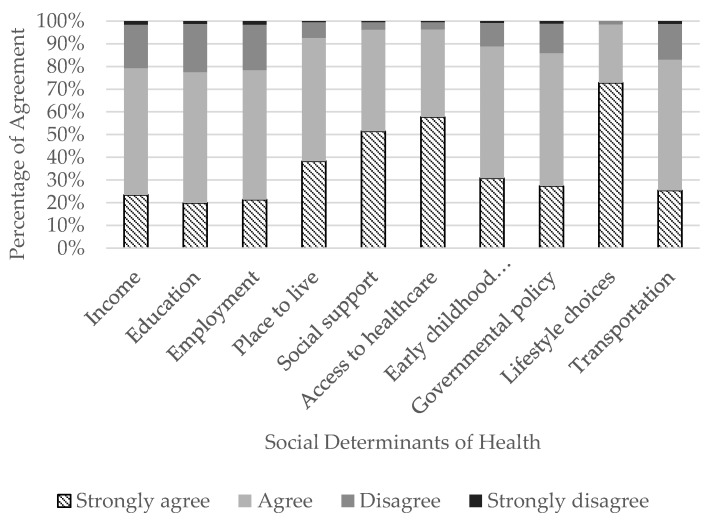
Athletic trainers’ perceptions of the influence social determinants of health have in athletic healthcare.

**Figure 3 ijerph-20-05602-f003:**
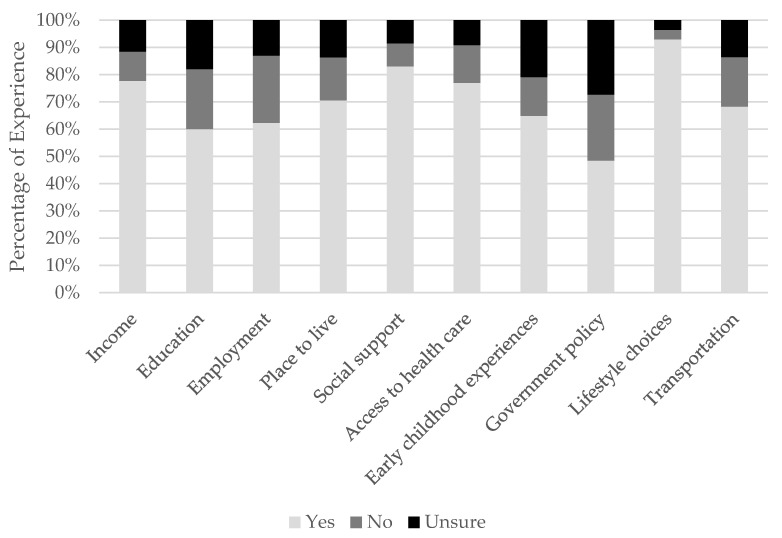
Athletic trainers’ experience managing social determinants of health when negatively influencing patient health and well-being.

**Table 1 ijerph-20-05602-t001:** Survey questions about athletic trainers’ perceptions of and experiences with social determinants of health (SDHs).

SDH	Questions Asked in the Survey
Access to quality and/or timely healthcare	(1) A person’s access to quality and/or timely health care services influences his/her health and well-being. (2) Have you ever managed a patient case in which the patient’s access to quality and/or timely health care services negatively influenced his/her health and well-being? (3) Globally, a patient’s access to quality and/or timely health care services is a concern in athletic healthcare.
Early childhood experiences	(1) A person’s early childhood experiences (such as type of parenting or upbringing and problems in the home) influence his/her health and well-being. (2) Have you ever managed a patient case in which the patient’s early childhood experiences negatively influenced his/her health and well-being? (3) Globally, a patient’s early childhood experiences are a concern in athletic healthcare.
Education	(1) A person’s level of education influences his/her health and well-being. (2) Have you ever managed a patient case in which the patient’s level of education negatively influenced his/her health and well-being? (3) Globally, a patient’s level of education is a concern in athletic healthcare.
Employment	(1) A person’s job or employment status influences his/her health and well-being. (2) Have you ever managed a patient case in which the patient’s job or employment status negatively influenced his/her health and well-being? (3) Globally, a patient’s job or employment status is a concern in athletic healthcare.
Government policies and programs	(1) Government policies and programs that affect health, social services, education, and economy influence a person’s health and well-being. (2) Have you ever managed a patient case in which government policies and programs negatively influenced the patient’s health and well-being? (3) Globally, government policies and programs are a concern in athletic healthcare.
Income	(1) A person’s income or the amount of money a person has influences his/her health and well-being. (2) Have you ever managed a patient case in which the patient’s income or amount of money negatively influenced his/her health and well-being? (3) Globally, a patients’ income or the amount of money a patient has is a concern in athletic healthcare.
Lifestyle choices	(1) A person’s lifestyle choices—what they eat, whether they smoke, how much alcohol they drink, and how much exercise they get—influence his/her health and well-being. (2) Have you ever managed a patient case in which the patient’s lifestyle choices negatively influenced his/her health and well-being? (3) Globally, a patient’s lifestyle choices—what they eat, whether they smoke, how much alcohol they drink, and how much exercise they get— are a concern in athletic healthcare.
Place to live	(1) Having a safe and affordable place to live influences a person’s health and well-being. (2) Have you ever managed a patient case in which the patient’s place to live negatively influenced his/her health and well-being? (3) Globally, a patient’s having safe and affordable place to live is a concern in athletic healthcare.
Social support	(1) Having the social support of others (such as family, friends, neighbors) who can help a person when in need influences his/her health and well-being. (2) Have you ever managed a patient case in which the patient’s lack of social support negatively influenced his/her health and well-being? (3) Globally, lack of social support for patients is a concern in athletic healthcare.
Transportation	(1) A person’s access to transportation (such as bus, taxi, personal vehicle, guardian ride) influences his/her health and well-being. (2) Have you ever managed a patient case in which the patient’s access to transportation negatively influenced his/her health and well-being? (3) Globally, a patient’s access to transportation is a concern in athletic healthcare.

## Data Availability

The data are not publicly available due to other studies in progress.

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
