# Peer review of "Athletic Trainers’ Perceptions of and Experience with Social Determinants of Health"

_ijerph, 2023, doi:10.3390/ijerph20085602_

Round 1

Reviewer 1 Report

Article  elaborates a current and significant topic using primary and secondary research. The article deals with the topic of the role that social determinants of health, focusing on a survey conducted among a special target group, athletic trainers. The article presents results based on a large number of sample surveys (N= 1694). The methodology and its presentation are correct and appropriate.

I professionally support the publication of this article. However, below I will formulate some criticisms, based on which I recommend that minor modifications be made.

1.     Lack of clarity in the research question: While the introduction provides some background information on social determinants of health (SDH) and their importance in patient outcomes, it doesn't clearly state what specific research question the study aims to answer. It mentions that the purpose of the study is to evaluate athletic trainers' perceptions and experiences with SDH, but it doesn't articulate what specific aspects of these perceptions and experiences the study will explore.

2.     Lack of specificity in the literature review: The introduction provides a brief overview of the importance of SDH in patient outcomes, but it doesn't provide specific examples or studies that have investigated these factors. This lack of specificity could weaken the argument for why studying athletic trainers' perceptions and experiences with SDH is important.

3.     Limited contextualization of the research: The introduction briefly mentions that the study will focus on athletic trainers' experiences, but it doesn't provide much context for why this population is important or unique in relation to SDH. Additionally, it doesn't acknowledge any potential limitations or biases in studying this population, such as the fact that athletic trainers may not be representative of all healthcare providers or patient populations.

4.     Lack of a clear hypothesis or research objective: While the introduction mentions the purpose of the study, it doesn't clearly state what the researchers hypothesize or hope to find. Without a clear hypothesis or objective, it can be difficult for readers to understand the significance of the research or how the study fits into the broader scientific literature on SDH.

5.     The limitations section could be more concise and focused on the specific limitations of the study, rather than including general information about the demographics of the participants.

Author Response

Article  elaborates a current and significant topic using primary and secondary research. The article deals with the topic of the role that social determinants of health, focusing on a survey conducted among a special target group, athletic trainers. The article presents results based on a large number of sample surveys (N= 1694). The methodology and its presentation are correct and appropriate.

I professionally support the publication of this article. However, below I will formulate some criticisms, based on which I recommend that minor modifications be made.

  1. Lack of clarity in the research question: While the introduction provides some background information on social determinants of health (SDH) and their importance in patient outcomes, it doesn't clearly state what specific research question the study aims to answer. It mentions that the purpose of the study is to evaluate athletic trainers' perceptions and experiences with SDH, but it doesn't articulate what specific aspects of these perceptions and experiences the study will explore.

We added a paragraph to the introduction to reflect on the data available from other providers and the need to obtain data from the AT community in order to help inform clinical strategies. The purpose statement uses perceptions of SDH to be broad enough to include all SDH. Table 1 provides a breakdown of each question asked and demonstrates that the questions covered the breadth of SDH. The purpose statement also says explore experiences with managing patient cases in which specific SDH may have been salient. We have removed “aspects of” to improve clarity. Specific SDH that were evaluated in the survey would be a lot to detail here, but are discussed in the methods section and table 1. By documenting ATs perceptions of SDH and exploring their experiences managing cases in which specific SDH may have been salient we hope to raise awareness of the importance of the SDH and to build the foundation for future work in this area. 

  1. Lack of specificity in the literature review: The introduction provides a brief overview of the importance of SDH in patient outcomes, but it doesn't provide specific examples or studies that have investigated these factors. This lack of specificity could weaken the argument for why studying athletic trainers' perceptions and experiences with SDH is important.

We have added a paragraph to show that other healthcare provider experiences with SDH and perceptions of SDH in practice have been studied. These data are helpful for those professions to better prepare them for supporting patients feeling the negative impact of SDH. Given the more specific patient population in athletic healthcare and the unique model of care delivery, similar explorations into clinician perceptions and experiences are needed to support athletic training practice.

  1. Limited contextualization of the research: The introduction briefly mentions that the study will focus on athletic trainers' experiences, but it doesn't provide much context for why this population is important or unique in relation to SDH. Additionally, it doesn't acknowledge any potential limitations or biases in studying this population, such as the fact that athletic trainers may not be representative of all healthcare providers or patient populations.

Thank you for this comment. We have strengthened this section by adding information to further highlight the uniqueness of the practice model and patient focus often seen in athletic healthcare. The delivery model (eg, daily patient interactions), practice setting (eg, often in school-aged patients), and patient focus (eg, high functioning patient populations) are different than traditional healthcare and we feel warrant specific exploration. It would be a mistake to generalize our findings to all healthcare providers or patient populations because they were not part of our sampling frame. The study was designed to address ATs and was not intended to represent all healthcare providers or patient populations, we address the lack of generalizability by adding to our limitations/future research section. It now includes “It is important to acknowledge that 65.2% of ATs who completed this survey were employed in either the secondary school, collegiate/university, or professional setting and thus are not representative of all ATs.”

  1. Lack of a clear hypothesis or research objective: While the introduction mentions the purpose of the study, it doesn't clearly state what the researchers hypothesize or hope to find. Without a clear hypothesis or objective, it can be difficult for readers to understand the significance of the research or how the study fits into the broader scientific literature on SDH.

Thank you for this comment. We’ve used a descriptive study design to explore our research questions and have not included comparative analyses. Because of the descriptive nature and no comparative analyses, specific hypotheses are not warranted. Descriptive studies are commonly undertaken to describe the basic features of a novel phenomenon. SDH have rarely been studied in athletic training populations, so we wished to explore and describe the construct phenomenology before proceeding to hypothesis construction. Without statistical tests, we do not have a hypotheses and believe there is still value in the descriptive nature of this work. We added the paragraphs described above to help clarify our purpose and emphasize the importance of this foundational work.

  1. The limitations section could be more concise and focused on the specific limitations of the study, rather than including general information about the demographics of the participants.

Thank you for this comment. Selection bias is a limitation, but we have reduced our comments on this limitation based on reviewer feedback. We explained the relationship of the demographics of the respondents to the lack of generalizability to all ATs. Thank you for allowing us to further clarify this.

Reviewer 2 Report

The authors are to be commended for their study that evaluated athletic trainers’ perceptions of social determinants of health (SDH) and their experience in treating patients affected by SDH. While the manuscript is well-structured and falls within the scope of the International Journal of Environmental Research and Public Health, there are significant revisions needed before publication can be recommended.

My primary concern with this study is the statistical analysis. The authors only relied on percentages, and while there are many statistical analyses that could be performed, at the very least, the authors should consider using a Chi-square test, which is the simplest method to compare percentages and check for any significance. Additionally, I recommend that the authors add a calculation of the sample size needed to achieve a meaningful statistical analysis.

Another important revision is to add a flowchart explaining the study protocol to help readers better understand the research design and methodology.

While the conclusions are clear and well-written, the practical implications of the study are missing, and I urge the authors to address this issue in both the abstract and the main text.

Finally, with the growing emphasis on research transparency among journals, editors, and reviewers, it is essential that the authors declare whether they used artificial intelligence (AI) at any stage of the manuscript writing process. I suggest that the authors mention this in the declaration section and cite the article by Dergaa et al. (2023) to learn more about AI declaration in academic writing and increase the transparency of scientific research.

Dergaa I, Chamari K, Zmijewski P, Ben Saad H. From human writing to artificial intelligence generated text: examining the prospects and potential threats of ChatGPT in academic writing. Biology of Sport. 2023;40(2):615-622. doi:10.5114/biolsport.2023.125623

In summary, the authors’ study has the potential to make an important contribution to the field of athletic training, but significant revisions are needed before it can be accepted for publication.

Author Response

Reviewer 2 Comments

The authors are to be commended for their study that evaluated athletic trainers’ perceptions of social determinants of health (SDH) and their experience in treating patients affected by SDH. While the manuscript is well-structured and falls within the scope of the International Journal of Environmental Research and Public Health, there are significant revisions needed before publication can be recommended.

My primary concern with this study is the statistical analysis. The authors only relied on percentages, and while there are many statistical analyses that could be performed, at the very least, the authors should consider using a Chi-square test, which is the simplest method to compare percentages and check for any significance. Additionally, I recommend that the authors add a calculation of the sample size needed to achieve a meaningful statistical analysis.

We acknowledge the reviewer’s concern about additional statistical analysis; however, we posed no hypotheses, and statistical tests are meaningless in the absence of hypotheses. We simply wanted to describe the basic attributes of athletic trainers’ perceptions of and experience with SDH. To stay true to the purpose of this descriptive study, we limited our analyses to descriptive statistics. As this descriptive study was not designed to test hypotheses, we wish to avoid performing tests which could imply statistical significance where comparisons were not part of the original research plan. Performing multiple tests can find statistical significance by chance and can be misleading. Thus, we hope to stick to our intention with a descriptive foundational piece. We are grateful for the review process where we have been able to make that intention more clear with the edits described above.

We appreciate the concern regarding sample size. However, with 1,000 respondents we feel comfortable that our aims were met. The literature supports use of data such as this for this foundational work where hypotheses are not being tested.

Another important revision is to add a flowchart explaining the study protocol to help readers better understand the research design and methodology.

Thank you for this suggestion. We chose not to include a visual such as a flow chart as it would only include number of people the survey was sent to, who accessed the survey, and who completed it, all of which we feel is clearly stated in the methods section. We will defer to the editors on whether a flow chart is needed.

While the conclusions are clear and well-written, the practical implications of the study are missing, and I urge the authors to address this issue in both the abstract and the main text.

Thank you for this comment and the reviewer 1 comments, we feel this has been addressed now with the addition of the background paragraph and statements described above.

Finally, with the growing emphasis on research transparency among journals, editors, and reviewers, it is essential that the authors declare whether they used artificial intelligence (AI) at any stage of the manuscript writing process. I suggest that the authors mention this in the declaration section and cite the article by Dergaa et al. (2023) to learn more about AI declaration in academic writing and increase the transparency of scientific research.

Dergaa I, Chamari K, Zmijewski P, Ben Saad H. From human writing to artificial intelligence generated text: examining the prospects and potential threats of ChatGPT in academic writing. Biology of Sport. 2023;40(2):615-622. doi:10.5114/biolsport.2023.125623

Thank you for sharing this citation. We have included a statement at the end of the manuscript stating that there was no use of AI at any time during this study.

In summary, the authors’ study has the potential to make an important contribution to the field of athletic training, but significant revisions are needed before it can be accepted for publication.

Thank you. We have incorporated your feedback and feel the manuscript is strengthened as a result.

Reviewer 3 Report

The authors could better substantiate their findings by analyzing the responses linked to the social and economic environment conditions where the coaches develop their activities, i.e. a better understanding of the determinants of health could be achieved by mapping the location of the coaches. although it is indicated that they are mostly working in secondary schools, the socio-demographic origin of the athletes with whom they interact could better explain the relevance of the determinants of health. 

In the literature there are some studies that show the relationship between the socioeconomic environments where individuals develop their lives and the cultural environment as elements directly related to lifestyle choices, this could expand the explanation of the results identified in the survey used.

It would be interesting for readers to better understand the relationship between the activity profiles of coaches, the socio-cultural and economic context in which they develop, to better understand the perception and estimates that transfer n the answers to the questions posed. 

Author Response

Reviewer 3 Comments

The authors could better substantiate their findings by analyzing the responses linked to the social and economic environment conditions where the coaches develop their activities, i.e. a better understanding of the determinants of health could be achieved by mapping the location of the coaches. although it is indicated that they are mostly working in secondary schools, the socio-demographic origin of the athletes with whom they interact could better explain the relevance of the determinants of health.

Thank you for this suggestion. This will be considered in future work, as this study did not ask questions of the athletic trainers that would allow for mapping specific location or the socio-demographic origin of the patients reflected on by participants in this study. While we collected the states and setting of employment for clinicians who participated (and reported for participant demographic information), those variables are unable to be connected with specific characteristics of patients reflected upon as part of this research.  

In the literature there are some studies that show the relationship between the socioeconomic environments where individuals develop their lives and the cultural environment as elements directly related to lifestyle choices, this could expand the explanation of the results identified in the survey used.

Thank you for this comment. In future studies of AT patients we will certainly explore the socioeconomic environments and their lifestyle choices. In this project where we surveyed ATs but not patients directly, we are only able to report the ATs observations and we can not overstep by commenting on any assumptions about the environment where the patients live and relating this to their lifestyle choices.

It would be interesting for readers to better understand the relationship between the activity profiles of coaches, the socio-cultural and economic context in which they develop, to better understand the perception and estimates that transfer n the answers to the questions posed.

Thank you for sharing this idea with us, although we are unsure what the reviewer is looking for here as this manuscript was exploring athletic trainers’ perceptions of social determinants of health and experiences managing patient cases where social determinants of health were negatively influencing their patient’s health. We did not collect any data related to coaches. We appreciate these thoughts and will consider future studies of coaches and evaluations of their socio-cultural and economic contexts.

Round 2

Reviewer 2 Report

I would like to thank the authors for submitting an improved version of the manuscript compared to the previous one. The changes made have significantly enhanced the clarity and overall quality of the article. At this stage, I believe the manuscript is ready for publication.